# Histopathologic Response Is a Positive Predictor of Overall Survival in Patients Undergoing Neoadjuvant/Perioperative Chemotherapy for Locally Advanced Gastric or Gastroesophageal Junction Cancers—Analysis from a Large Single Center Cohort in Germany

**DOI:** 10.3390/cancers12082244

**Published:** 2020-08-11

**Authors:** Rebekka Schirren, Alexander Novotny, Helmut Friess, Daniel Reim

**Affiliations:** TUM School of Medicine, Department of Surgery, Ismaninger Strasse 22, 81675 Munich, Germany; rebekka.schirren@tum.de (R.S.); alexander.novotny@tum.de (A.N.); helmut.friess@tum.de (H.F.)

**Keywords:** gastric/gastroesophageal cancer, perioperative chemotherapy, overall survival, relapse-free survival

## Abstract

There is conflicting evidence regarding the efficacy of neoadjuvant/perioperative chemotherapy (NCT) for gastro-esophageal cancer (GEC) on overall survival. This study aimed to analyze the outcomes of multimodal treatments in a large single center cohort. We performed a retrospective analysis of patients treated with NCT, followed by intended curative oncological surgery for locally advanced gastric cancer. Uni- and multivariate regression analysis were performed to identify the predictors of overall survival. From over 3000 patients, 702 eligible patients were analyzed. In the univariate analysis clinical stage, application of preoperative PLF, requirement of surgical extension, UICC-stage, grading, R-status, Lauren histotype, and HPR were the prognostic survival factors. In multivariate analysis PLF regimen, UICC-stages, R-status, Lauren histotype, and histopathologic regression (HPR) were significant predictors of overall survival. Overall HPR-rate was 26.9%. HPR was highest in the cT2cN0 stage (55.9%), and lowest in the cT3/4 cN+ stage (21.6%). FLOT demonstrated the highest HPR (37.5%). Independent predictors for HPR were the clinical stage and grading. Kaplan Meier analyses demonstrated significant survival benefits for the responding patients (*p* < 0.0001). HPR after NCT was an important prognostic factor to predict overall survival for locally advanced GEC. FLOT should be the preferred regimen in patients undergoing NCT ahead of surgery.

## 1. Introduction

Gastric cancer belongs to the most common malignant diseases worldwide, with the highest incidence in Eastern Asia [1]. Despite decreasing incidence in the West, it remains a therapeutic challenge. In the Western hemisphere, gastric malignancy is often diagnosed at an advanced stage and in contrast to Eastern Asia, it is preferably located in the proximal third of the stomach or the gastro-esophageal junction (GEJ) [2]. Hence, multimodal treatment concepts were introduced, after demonstrating outcome benefits in randomized controlled trials [3,4,5]. Nevertheless, there is still conflicting evidence, that perioperative chemotherapy might not be effective for all patients, especially those with non-cardia gastric cancer and poorly cohesive type gastro-esophageal cancer [6,7,8]. New chemotherapeutic regimens were introduced into clinical practice in the last few years, the most promising being the FLOT regimen (Fluorouracil, Leucovorin, Oxaliplatin, and Docetaxel), which demonstrated higher histopathological regression rates and which was shown to be an independent prognostic factor for overall and disease-free survival [9]. The aim of this analysis was to evaluate oncological outcomes and predictors of perioperative/neoadjuvant chemotherapy in a large German single center cohort.

## 2. Results

### 2.1. Patient Data

During the designated period from 1987 to 2014, over 3000 patients were treated for gastric cancer at the Surgical Department of TUM, from which 894 patients underwent intended neoadjuvant/perioperative chemotherapy. Patients undergoing R2 resection (*n* = 47) and the metastatic patients (*n* = 145) were excluded from the analysis. Finally, 702 patients fulfilled the inclusion criteria and were eventually included in this analysis. Most patients were male (75%) and the tumors were predominantly located at the gastro-esophageal junction (68%). The most frequently applied chemotherapeutic regimen was PLF (50%). Two-thirds of the patients required surgical extension for complete tumor removal, mostly extending to the distal esophagus. The overall morbidity rate was 26%. The median number of dissected lymph nodes was 29 [range 5–128]. A total of 72% of all patients demonstrated ypT3/ypT4 tumors and 56% of patients had lymph node metastases. Most patients (73%) had poorly differentiated (G3/G4) histology. Almost half of the cases demonstrated Lauren intestinal-type histology (48%), followed by diffuse-type (25%). R0-resections were achieved in 87%, and almost 27% of patients revealed a histopathological response (Becker 1a/Becker 1b) [10] to preoperative chemotherapy. Moderate response (Becker 2 (10–50% remaining viable tumor cells) was detected in 29% and poor response (Becker 3 (>50% remaining viable tumor cells) was found in 44%. The representative histopathological slides are shown in Appendix A, for each histopathological response grade. The extensive baseline characteristics are depicted in Table 1.

Median follow-up was 56 months (range 2–269 months), comprising of 59.5 months [range 12–69 months] for survivors and 18 months (range 1–216) months for deceased patients. During the follow-up period, 346 patients (49.3%) died, the five-year survival rate was 46%, the ten-year survival rate was 32% (*p* = 0.003). Median survival for the histopathologic responders was 216 months and 36 months for non-responders (*p* < 0.0001). The five- and ten-year survival probabilities were 70%/60% for responders and 40%/29% for non-responders, respectively. Kaplan Meier analyses demonstrated a statistically significant survival benefit for responders, compared to non-responders (Figure 1). No survival benefit was detected for the intermediate responders (Becker 2), compared to the non-responding patients ((Becker 3), *p* = 0.155) (Figure 2).

### 2.2. Predictors of Overall Survival

Univariate regression analysis revealed clinical stage, application of preoperative PLF, requirement of surgical extension, UICC-stage, grading, R-status, Lauren histotype (intestinal and diffuse types), and histopathologic response to be significantly related to postoperative survival (Table 2).

The multivariate analysis demonstrated that PLF regimen, UICC-stages, R-status, Lauren histotype (intestinal and diffuse), and histopathologic response were significantly and independently related to postoperative survival (Table 3).

### 2.3. Histopathologic Response

Histopathologic response as defined by grade 1a/1b, according to the Becker classification, was evaluated postoperatively, as described above. The overall histopathologic response rate was 26.9%. Early clinical stages (cT2-4/cN0) revealed higher histopathologic response rates than advanced stages with lymph node involvement (33–55% vs. 21–33%, *p* < 0.001). With regards to the chemotherapeutic regimens, FLOT revealed the highest histopathologic response rate (37.5%), followed by Taxol+PLF (35.1%), PLF (26.1%), ECF (23.4%), and lastly OLF (17.4%). However, this result was not statistically significant (*p* = 0.103). The response rates varied by the UICC stage: In UICC stage I, there were 71.7% responders (114 of 159 patients), in UICC II there were 19.9% (50/251), and in UICC III, there were 8.6% responders (25/265) (*p* < 0.0001). The proportion of histopathologic responders was higher in the AEG-group than in the non-AEG group (29.8% vs. 20.9%, *p* < 0.0001). Results are shown in Table 4.

#### Predictors of HPR

Clinical factors predicting whether patients were more likely to respond to chemotherapy were evaluated by multivariate regression analysis. In the univariate model, tumor location, gender, clinical stage, intestinal Lauren histotype, and grading were the predictors for histopathologic regression. In the multivariate model, only the clinical stage and grading were significantly related to the histopathologic response. The extensive results are shown in Table 5.

## 3. Discussion

This analysis of a large single center cohort demonstrated that neoadjuvant/perioperative chemotherapy results in survival benefit only in patients who demonstrate histopathologic response, as demonstrated by Kaplan Meier and multivariate regression analyses. Histopathologic response was defined when there was either no viable or less than 10% viable tumor cells, in relation to the detectable tumor bed. Patients demonstrating intermediate response according to the Becker classification revealed no benefits over those patients not responding to neoadjuvant/perioperative chemotherapy. This analysis found that only a little more than a quarter of patients respond to neoadjuvant/perioperative chemotherapy, which leads to the notion that almost three-quarters of all patients do not benefit from neoadjuvant/perioperative chemotherapy at all. It remains elusive if these patients were not possibly even been harmed by the ineffective treatment ahead of surgery. Interestingly, histopathologic response rates differed, depending on the chemotherapeutic regimen applied. Among these, three substance-based therapies like FLOT and Taxol-PLF were the most effective regarding response rates. However, this effect was not statistically significant, because the case numbers were too small to draw definitive conclusions. The phase III study on FLOT demonstrated promising results, which need to be proven in clinical practice in the near future [9]. Another remarkable result was that the early clinical stages (cT1/cT2) and patients without clinical detection of lymph node involvement (cN0) revealed high HPR rates. The reasons for this fact are difficult to determine. The elusive reasons might be simple understaging of the real situation or favorable tumor biology in earlier stages, when the cancer does not reach its metastatic potential and responsiveness to chemotherapy is higher than that in later stages. Further reasons for reduced histopathologic responsiveness might also be poor or undifferentiated tumor grading, which is a statistically significant predictor for worse response to neoadjuvant chemotherapy [6,11]. Another factor might be Lauren differentiation. In this analysis, almost half of the patients demonstrated intestinal types, which are considered to be more responsive to chemotherapy than Lauren diffuse types. There is an ongoing discussion about chemotherapy responsiveness related to the histopathologic subtype [11,12]. Lauren diffuse types also incorporate signet ring cell cancers, poorly cohesive cancers with signet ring cells, and poorly cohesive cancer without signet ring cell differentiation. Previously, it was found that signet ring cell differentiation might be related to poor responsiveness [8,12]. However, these analyses are difficult to compare because of different “signet ring cell” classifications. A standardized approach was taken by an expert group, however, this was not yet evaluated for patients undergoing neoadjuvant chemotherapy and was not yet validated in an international setting [13]. Nonetheless, in this analysis, the Lauren diffuse type differentiation resulted in a 40% higher risk of death compared to the intestinal type, but the intestinal type was no independent predictor of HPR. Further study on this fact is required to elucidate the influence of Lauren histotype on histopathologic response to neoadjuvant chemotherapy, especially in a standardized and comparable way. In contrast to previously published reports, this analysis could not confirm the influence of tumor localization on histopathologic response [6,14]. Multivariate analyses on both the overall survival and the histopathologic response prediction were not able to demonstrate an effect related to the tumor site (cardia vs. non-cardia).

Several limitations of this analysis were evident, besides its single-center character and retrospective design. The inclusion period covers a long period of time of thirty years. During this period, surgical techniques might not have changed too much (except minimal invasive technologies) but peri-operative care and management of postoperative complications certainly has, which might have influenced oncological outcomes over time. These innovations were not included in this analysis, because the data were not available. Besides this, chemotherapeutic regimens changed over time, influenced by published results from randomized controlled trial [3,5,9,15]. Further, no toxicity data of the chemotherapeutic treatments were available in the database to analyze if dose or cycle reductions were necessary and might have influenced histopathologic response rates. Further, the newest innovation was the introduction of FLOT as a new standard of perioperative chemotherapy, which was underrepresented in this analysis, due to a small number of patients being eligible for analysis [9]. Besides this, comparisons of the regimens might be erroneous as the MAGIC and FLOT protocols consist of additional postoperative (adjuvant) chemotherapy, whereas the other protocols only consist of preoperative (neoadjuvant) treatment [3,9]. Therefore survival outcome comparisons might be biased because the effect of the adjuvant part could not be properly evaluated. Further, recurrence rates and recurrence-free survival data were not analyzed because they were not available from the present database. Lastly, the frequency and quality of comorbidities was not analyzed, because the data were not available in the database. Certainly these comorbidities might have influenced dose adaptions during neoadjuvant chemotherapy and might have influenced not only the oncologic but also surgical outcomes, which represents a substantial limitation of this retrospective analysis.

Certainly, generalizability of the results presented here is limited, due to the fact that neoadjuvant/perioperative chemotherapy is part of clinical practice only in Europe, whereas the concept of primary resection, followed by adjuvant chemotherapy is practiced predominantly in Eastern Asia, and the concept of (neo-)adjuvant chemo-radiation is commonly accepted in the US [3,5,15,16,17]. Besides this, cardia cancers are often treated by neoadjuvant chemo-radiation, since publication of the CROSS study [4,16]. These practices were not represented in this analysis, which limits the general applicability of the present results.

## 4. Materials and Methods

### 4.1. Patients

Data from patients who underwent curative surgery for gastroesophageal cancer at the Surgical Department of TUM School of Medicine from 1987 to 2017 were extracted from a prospectively documented database. Data were obtained from the medical records and transferred to the institutional databases, as soon as the patients were discharged from inpatient hospital care. The inclusion criteria for this analysis were—histologically proven gastroesophageal cancer (Siewert type II/III, all non-cardia cancers) staged cT2-cT4cN_any_ undergoing neoadjuvant/perioperative chemotherapy, after a multidisciplinary team review. Exclusion criteria were—Siewert type I, metastatic disease, hospital mortality within 30 days, loss of follow-up within a 60 months period and macroscopic residual cancer after surgery (R2). Neoadjuvant/perioperative treatment consisted of either preoperative two cycle—cisplatin or oxaliplatin/leucovorin/5-FU (PLF/OLF), only or perioperative three cycles of ECX/ECF (MAGIC) or perioperative four cycles FLOT [3,9]. All surgical procedures were performed according to the Japanese guidelines for GC treatment, including standardized D2-lymphnode dissection [18]. In case of GE junction cancer (Siewert type II and III), the surgical procedure was extended to the distal esophagus. All patients received intraoperative frozen sections for the oral resection margin to confirm R0 resection. Circumferential and aboral resection margins were not determined intraoperatively on a routine basis. All resected specimens were examined by one or two specialized pathologists, classified according to the TNM-classification, and staged according to the UICC-recommendations (8th edition) [19]. Histopathologic response was graded according to the Becker classification. Patients 0–10% remnant viable tumor cells within the tumor area were graded as histopathologic responders (Becker 1a/1b), whereas all other patients (Becker 2 (10–50% remnant viable tumor cells) and Becker 3 (>50% remnant viable tumor cells)) were graded as histopathologic non-responders [10]. Following oncologic surgery, all patients were followed up every six to twelve months, in an outpatient department (Roman Herzog Comprehensive Cancer Center), over the next five years, using EGD and CT scans, according to the institutional protocol.

Only deceased or surviving patients with complete follow-up of at least 60 months were included in this analysis. Survival was computed from the day of surgery. The dataset consisted of patients’ gender, age, location (upper, middle, lower third), clinical stages (cT2N0, cT1/cT2cN+, cT3/cT4cN0, cT3/cT4N+), type of chemotherapeutic regimen applied (PLF, OLF, Taxol+PLF, ECF/ECX, FLOT, modified platin-based CTx), type of surgery (esophagectomy, transhiatal gastrectomy, gastrectomy, subtotal gastrectomy), type of required extension (none, luminal/transhiatal, splenectomy, colon, pancreas, others), number of dissected lymph nodes, postoperative complications (none, Clavien–Dindo Grade I/II, and III/IV), pT- (pT0/pT1a/pT1b/pT2/pT3/pT4a/pT4b), pN-(pN0/pN1/pN2/pN3a/pN3b), and UICC-stages (UICC-0/-IA/-IB/-IIA/-IIB/-IIIA/-IIIB/-IIIC), grading (G1/2, G3/4), R-status (R0/R1), Lauren histotype (intestinal, diffuse, mixed, non-classified), and follow-up period with survival status. Institutional Review Board (IRB)-approval for this study was obtained according to the local guidelines (IRB Registration: 364/20 S).

### 4.2. Statistical Analysis

Wilcoxon and chi-square tests were used to compare the continuous and categorical clinical characteristics. Overall survival (OS) was graphed using empirical Kaplan-Meier curves with differences in 5-year survival rates among the patient groups evaluated using the log-rank test. Associations between prognostic factors, and survival were estimated by uni- and multivariate Cox proportional-hazards regression analysis. Histopathologic response predictors were evaluated by multivariate regression analysis. All variables were included in the multivariate model to rule out possible confounding for both outcomes. All statistical tests were performed at the two-sided 0.05 level of significance. Statistical analyses were performed using SPSS-software (Version 24, IBM Inc., Armonk, NY, USA).

## 5. Conclusions

In conclusion, histopathologic response after neoadjuvant chemotherapy is an important prognostic factor to predict overall survival for locally advanced gastro-esophageal cancer. FLOT should be the preferred therapeutic regimen in patients undergoing neoadjuvant/perioperative chemotherapy ahead of surgery. Further research should focus on the early detection of patients not responding well to multimodal treatment.

## Figures and Tables

**Figure 1 cancers-12-02244-f001:**
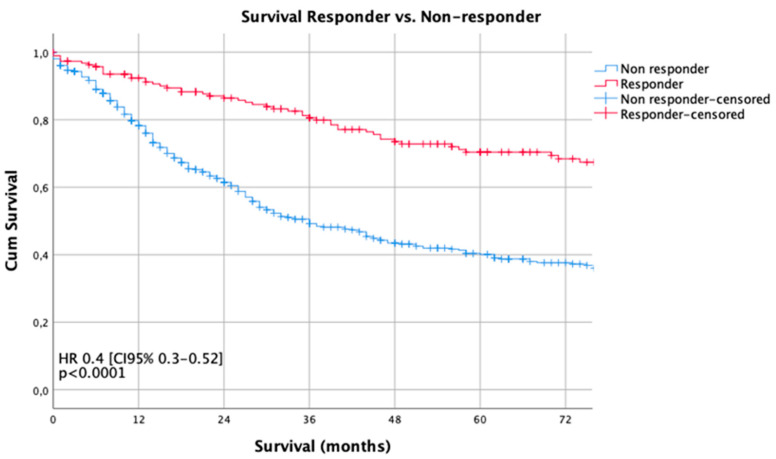
Survival curves according to histopathologic regression (HPR). The Kaplan-Meier method and the log-rank test were used to compare the estimated survival by histopathologic responders and non-responders.

**Figure 2 cancers-12-02244-f002:**
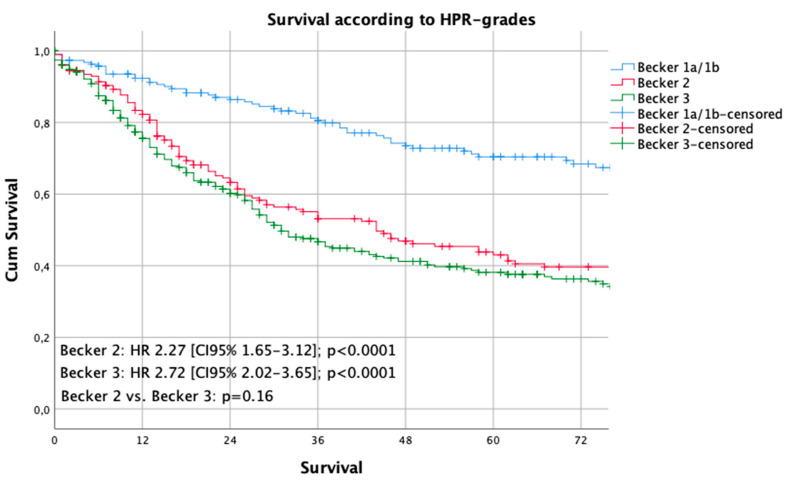
Survival curves according to the Becker grades. The Kaplan-Meier method and the log-rank test were used to compare the estimated survival by each Becker-stage.

**Table 1 cancers-12-02244-t001:** Baseline characteristics.

Characteristics	*n*	%
Gender		
Female	172	24.50
Male	530	75.50
Age (years) *	58.8+/−11.5 (range 3–83 years)	
<70	590	84.05
>70	112	15.95
Localization		
Siewert II/III ^#^	477	67.95
Middle	111	15.81
Distal	88	12.54
Total	26	3.70
Clinical Staging ^$^		
cT2 cN0	56	7.98
cT1/cT2 cN+	57	8.12
cT3/cT4 cN0	102	14.53
cT3/cT4 cN+	487	69.37
Type of chemotherapy ^&^		
PLF	351	50.00
OLF	70	9.97
Taxol+PLF	57	8.12
ECF/ECX	64	9.12
FLOT	56	7.98
Modified platin based CTx	104	14.81
Type of Surgery		
Esophagectomy	147	20.94
Transhiatal ext. Gastrectomy	326	46.44
Total gastrectomy	191	27.21
Subtotal gastrectomy	38	5.41
Surgical extension		
None	238	33.90
Luminal/transhiatal	288	41.03
Splenectomy	19	2.71
Colon	5	0.71
Pancreas	18	2.56
Others	134	19.09
Dissected LN [Median]	29 (Range 5–218)	
<=25	232	33.05
>25	470	66.95
Complications ^?^		
None	515	73.36
CD I/II	84	11.97
CD III-V	103	14.67
pT ^!^		
pT0/is	35	4.99
pT1a	22	3.13
pT1b	50	7.12
pT2	88	12.54
pT3	331	47.15
pT4a	148	21.08
pT4b	28	3.99
pN ^!^		
pN0	306	43.59
pN1	130	18.52
pN2	109	15.53
pN3a	106	15.10
pN3b	51	7.26
UICC ^!^		
UICC 0	32	4.56
UICC IA	58	8.26
UICC IB	69	9.83
UICC IIA	126	17.95
UICC IIB	125	17.81
UICC IIIA	97	13.82
UICC IIIB	134	19.09
UICC IIIC	61	8.69
Grading		
G1/G2	191	27.21
G3/G4	511	72.79
R		
R0	615	87.61
R1	87	12.39
Lauren histotype		
Intestinal	339	48.29
Diffuse	177	25.21
Mixed	92	13.11
Not classified	94	13.39
Histopathologic Response		
Becker Ia/Ib	189	26.92
Becker II	202	28.77
Becker III	311	44.30

* Mean ± standard deviation; # GE-Junction cancer according to Siewert classification; $ cT1 = Mucosa/Submucosa; cT2 = Muscularis propria; cT3 = Serosa; cT4 = Adjacent organs; cN0 = no lymph nodemetastasis detected during staging, cN+ = locoregional lymph node metastasis evident during staging; & PLF = 2 cycles preOP; OLF; 2 cycles preOP; Taxol/PLF 2 cycles preOP, ECF/ECX = 3 cycles preOP+3cycles postOP; FLOT = 4 cycles preOP and 4 cycles postOP; ? According to Clavien Dindo classification; ! UICC 8th edition.

**Table 2 cancers-12-02244-t002:** Univariate analysis of predictors for overall survival.

Univariate	HR	CI95% Lower	CI95% Upper	*p*
Gender ^!^	1.19	0.92	1.53	0.190
Age (>70y)	1.23	0.93	1.64	0.150
Localization ^§^	1.26	0.99	1.59	0.060
cT2 cN0 ^$^	1.00			0.032
cT3/4 cN0	1.65	0.99	2.74	**0.050**
cT1/2 cN+	1.74	1.00	3.02	**0.050**
cT3/4 cN+	1.90	1.23	2.93	**0.004**
PLF ^$^	1.00			**0.007**
OLF	1.20	0.83	1.72	0.335
MAGIC	0.89	0.58	1.37	0.594
FLOT	0.39	0.16	0.96	0.040
PLF-Taxol	0.79	0.53	1.17	0.241
Other	1.46	1.10	1.94	**0.008**
Esophagectomy ^$^	1.00			0.052
Extended gastrectomy	1.17	0.88	1.56	0.274
Gastrectomy	0.88	0.63	1.22	0.430
Subtotal Gastrectomy	0.67	0.37	1.22	0.192
Surgical Extension	1.37	1.08	1.73	**0.009**
LN dissected (>25/<25)	1.02	0.81	1.28	0.870
Complication (any) ^+^	1.21	0.96	1.53	0.110
UICC I ^$^	1.00			**0.000**
UICC II	2.74	1.87	4.01	**0.000**
UICC III	5.48	3.80	7.90	**0.000**
G1/2 vs G3/4	1.61	1.24	2.09	**0.000**
R1 vs. R0	2.43	1.84	3.20	**0.000**
Lauren intestinal ^$^	1.00			**0.001**
Lauren diffuse	1.51	1.18	1.93	**0.001**
Lauren mixed	0.97	0.68	1.39	0.884
Lauren not classified	0.79	0.56	1.11	0.175
HPR (Y/N)	0.39	0.30	0.52	**0.000**

HR = Hazard Ratio, CI95% lower: 95% Confidence Interval lower boundary, CI95% upper: 95% Confidence Interval upper boundary, *p* = *p*-value, HPR=histopathologic response according to Becker; ^$^ cT1 = Mucosa/Submucosa; cT2 = Muscularis propria; cT3 = Serosa; cT4 = Adjacent organs; cN0: no lymph node metastasis detected during staging, cN+: locoregional lymph node metastasis evident during staging; ! male vs. female; ^§^ GE-junction vs. distal gastric cancer; $ categorical variable, first value is reference (=1.00); + Any complication vs. no complication. Bold variables are considered statistically significant.

**Table 3 cancers-12-02244-t003:** Multivariate analysis of predictors for overall survival.

Multivariate	HR	CI95% Lower	CI95% Upper	*p*
Gender ^!^	1.22	0.93	1.60	0.154
Age (>70y)	1.08	0.79	1.46	0.635
Localization ^§^	1.20	0.71	2.02	0.492
cT2 cN0 ^$^	1.00			0.550
cT3/4 cN0	1.29	0.76	2.20	0.354
cT1/2 cN+	0.90	0.51	1.61	0.733
cT3/4 cN+	1.05	0.66	1.68	0.830
PLF ^$^				**0.033**
OLF	1.01	0.69	1.47	0.980
MAGIC	0.94	0.61	1.44	0.761
FLOT	0.53	0.21	1.29	0.161
PLF-Taxol	0.74	0.49	1.11	0.140
Other	1.47	1.09	1.99	**0.013**
Esophagectomy ^$^	1.00			0.261
Extended gastrectomy	1.17	0.87	1.57	0.298
Gastrectomy	0.79	0.44	1.43	0.440
Subtotal Gastrectomy	0.61	0.27	1.37	0.228
Surgical Extension	1.00	0.71	1.41	0.992
LN dissected (>25/<25)	0.82	0.64	1.04	0.104
Complication (any) ^+^	1.17	0.91	1.49	0.222
UICC I ^$^	1.00			**0.000**
UICC II	2.07	1.35	3.16	**0.001**
UICC III	3.98	2.58	6.13	**0.000**
G1/2 vs. G3/4	1.21	0.89	1.65	0.234
R1 vs. R0	1.50	1.11	2.02	**0.009**
Lauren intestinal ^$^	1.00			**0.002**
Lauren diffuse	1.40	1.03	1.91	**0.034**
Lauren mixed	0.91	0.62	1.34	0.641
Lauren not classified	0.66	0.45	0.96	0.031
HPR (Y/N)	0.71	0.51	0.99	**0.045**

HR = Hazard Ratio, CI95% lower: 95% Confidence Interval lower boundary, CI95% upper: 95% Confidence Interval upper boundary, *p* = *p*-value, HPR = histopathologic response according to Becker; ^$^ cT1 = Mucosa/Submucosa; cT2 = Muscularis propria; cT3 = Serosa; cT4 = Adjacent organs; cN0: no lymph node metastasis detected during staging, cN+: locoregional lymph node metastasis evident during staging; ! male vs. female; ^§^ GE-junction vs. distal gastric cancer; $ categorical variable, first value is reference (=1.00); + Any complication vs. no complication. Bold variables are considered statistically significant.

**Table 4 cancers-12-02244-t004:** HPR rates according to the clinical factors.

Variable	NR	%	Responder	%	Total	*p*
cT2 cN− ^$^	25	44.6	31	**55.4**	56	**<0.001**
cT3/4 cN−	68	66.7	34	33.3	102	
cT1/2 cN+	38	66.7	19	33.3	57	
cT3/4 cN+	382	78.4	105	21.6	487	
Total	513	73.1	189	26.9	702	
PLF^&^	263	73.9	93	26.1	356	0.103
OLF	58	82.9	12	17.1	70	
MAGIC	49	76.6	15	23.4	64	
FLOT	35	62.5	21	**37.5**	56	
PLF-Taxol	37	64.9	20	35.1	57	
Other	71	71.7	28	28.3	99	
Total	513	73.1	189	26.9	702	
Non-AEG^#^	178	79.11	47	20.89	225	
AEG	335	70.23	142	29.77	477	
Total	513	73.1	189	26.9	702	
UICC I ^!^	45	28.30	114	71.70	159	***p* < 0.0001**
UICC II	201	80.08	50	19.92	251	
UICC III	267	91.44	25	8.56	292	
Total	513	73.1	189	26.9	702	

NR = Non-responder according to Becker classification, Responder = responder according to the Becker classification, *p* = *p*-value. ^$^ cT1 = Mucosa/Submucosa; cT2 = Muscularis propria; cT3 = Serosa; cT4 = Adjacent organs; cN0: no lymph node metastasis detected during staging, cN+: locoregional lymph node metastasis evident during staging; ^&^ PLF: 2 cycles preOP; OLF; 2 cycles preOP; Taxol/PLF 2 cycles preOP, ECF/ECX: 3 cycles preOP+3cycles postOP; FLOT: 4 cycles preOP and 4 cycles postop; ^#^ GE-Junction cancer according to Siewert classification; ^!^ UICC 8th edition. Bold variables are considered statistically significant.

**Table 5 cancers-12-02244-t005:** Uni-/multivariate analysis for the predictors of HPR.

Univariate	OR	CI95% Lower	CI95% Upper	*p*
Localization ^§^	1.61	1.10	2.34	**0.01**
Gender ^!^	1.53	1.02	2.32	**0.04**
Age (>70y)	0.89	0.56	1.41	0.62
cT2 cN0 ^$^	1.00			**<0.001**
cT3/4 cN0	0.40	0.21	0.79	**0.01**
cT1/2 cN+	0.40	0.19	0.86	**0.02**
cT3/4 cN+	0.22	0.13	0.39	**0.00**
Lauren type (intest. vs. other)	1.77	1.14	2.46	**0.01**
Grading (G1/2 vs. G3/4)	0.34	0.24	0.49	**<0.001**
MULTIVARIATE	OR	CI95% lower	CI95% upper	*p*
Localization ^§^	1.44	0.83	2.52	0.20
Gender ^!^	1.70	0.92	3.13	0.09
Age (>70y)	0.57	0.29	1.12	0.10
cT2 cN0 ^$^	1.00			**<0.001**
cT3/4 cN0	0.20	0.03	1.18	0.08
cT1/2 cN+	0.58	0.08	4.24	0.59
cT3/4 cN+	0.12	0.02	0.65	**0.01**
Lauren type (intest. vs. other)	1.30	0.75	2.22	0.35
Grading (G1/2 vs. G3/4)	0.47	0.27	0.82	**0.01**

OR = Odds Ratio, CI95% lower: 95% Confidence Interval lower boundary, CI95% upper: 95% Confidence Interval upper boundary, *p* = *p*-value, HPR = histopathologic response according to Becker; ^$^ cT1 = Mucosa/Submucosa; cT2=Muscularis propria; cT3 = Serosa; cT4 = Adjacent organs; cN0: no lymph node metastasis detected during staging, cN+: locoregional lymph node metastasis evident during staging; ^§^ GE-junction vs. distal gastric cancer; ! male vs. female; $ categorical variable, first value is reference (=1.00). Bold variables are considered statistically significant.

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
