# Peer review of "Histopathologic Response Is a Positive Predictor of Overall Survival in Patients Undergoing Neoadjuvant/Perioperative Chemotherapy for Locally Advanced Gastric or Gastroesophageal Junction Cancers—Analysis from a Large Single Center Cohort in Germany"

_cancers, 2020, doi:10.3390/cancers12082244_

Round 1

Reviewer 1 Report

In the present manuscript the Authors have performed a retrospective study of 702 gastro-esophageal cancer patients treated with neoadjuvant/perioperative chemotherapy to analyze the outcomes of multimodal treatments. In univariate analysis clinical stage, application of preoperative PLF, requirement of surgical  extension, stage, grading, R-status, Lauren histotype and HPR appeared as prognostic survival factors. In multivariate analysis PLF regimen, stage, R-status, Lauren histotype and histopathologic regression emerged as significant predictors of overall survival.

In my opinion, the overall level of the paper is very good structured: it is concisely written and some important considerations are highlighted.

The discussion and conclusions sections offer useful information for the readers.

However,  I have only a minor suggestion to make the manuscript more interesting:

  1. Please add some microscopic images of histopathologic response

Author Response

We kindly appreciate the reviewers positive comments. The additional images were included in the manuscript.

Reviewer 2 Report

This is a well conducted retrospective analysis of over 700 patients with oesophagogastric cancer treated with neoadjuvant chemotherapy. The main message from this study confirms that histopathological regression predicts for overall survival, which is already accepted in the oncology community. The manuscript is carefully written and with appropriate discussion of the limitations of this study in the last two paragraphs of the conclusions section.

Author Response

We kindly appreciate the reviewers positive comments and are happy to have met the reviewer´s expectations.

Reviewer 3 Report

The manuscript by Schirren and co-workers analyze the predictive factors on GEC patients with perioperative CT. The manuscript is well written and presented. Certainly, the authors indicate several limitations in the results, but still the provided information shows important aspects of the different treatment regiments in this type of patients. Nevertheless, there are several issues that need to be addressed before considering this manuscript for publication.

Major concerns

  1. What about co-morbidities. Did the authors review or analyzed something about it? It is important to mention this part either in the results and the discussion sections.
  2. The Kaplan-Meier curves look poor, for instance the authors indicate that the median survival for responders is 216 months which is not showed in the graph. Authors must include the censored events as well as the hazard ratios and confidence intervals in order to complete the displayed information.

Minor concerns

  1. Line 42, define FLOT
  2. Line 43, it is not clear how the FLOT regimen is an independent factor. Authors need to specify whether they refer to OS, DFS, PFS, etc.
  3. Line 57, references indicated are 5-128, must be corrected.
  4. Please indicate the age range, this will provide useful information for the readers

Author Response

  1. Comorbidities were not analyzed. Unfortunately we cannot provide data on comorbidities, because they were not documented in the database. We included a statement in the discussion section that this is a substantial limitation of this analysis.
  2. The Kaplan Meier curves were updated according to the reviewer´s suggestions and the requested data is now displayed.
  3. FLOT: Fluorouracil, leucovorin, oxaliplatin, docetaxel. This was included in the revised manuscript (line 43).
  4. The FLOT trial revealed that the FLOT was effective to improve overall and disease free survival. This information was added in the manuscript (line 45).
  5. This is not a reference. This is the range of lymph nodes dissected. The term “range” was included in the brackets (line 59).
  6. The age range was 23-83 years. This was changed in the table 1 (age), line 68)